# Are Healthcare Workers Infected with SARS-CoV-2 at Home or at Work? A Comparative Prevalence Study

**DOI:** 10.3390/ijerph191912951

**Published:** 2022-10-10

**Authors:** Shadi Zahran, Ran Nir-Paz, Ora Paltiel, Chen Stein-Zamir, Yonatan Oster

**Affiliations:** 1Department of Internal Medicine, Hadassah Hebrew University Medical Center, Jerusalem 9112001, Israel; 2Faculty of Medicine, Hebrew University of Jerusalem, Jerusalem 9190401, Israel; 3Department of Clinical Microbiology and Infectious Diseases, Hadassah Hebrew University Medical Center, Jerusalem 9112001, Israel; 4Faculty of Medicine, Braun School of Public and Community Medicine, Hebrew University of Jerusalem, Jerusalem 9190401, Israel; 5Jerusalem District Health Office, Ministry of Health, Jerusalem 9134302, Israel

**Keywords:** SARS-CoV-2, COVID-19, healthcare workers, neighborhood, location, occupational risk

## Abstract

Prior to the widespread use of vaccinations, healthcare workers (HCWs) faced the double burden of caring for unprecedented numbers of critically ill COVID-19 patients while also facing the risk of becoming infected themselves either in healthcare facilities or at home. In order to assess whether SARS-CoV-2-positivity rates in HCWs reflected or differed from those in their residential areas, we compared the SARS-CoV-2-positivity rates during 2020 among HCWs in Hadassah Hebrew University Medical Centers (HHUMC), a tertiary medical center in Jerusalem, Israel, to those of the general population in Jerusalem, stratified by neighborhood. Additionally, we compared the demographic and professional parameters in every group. Four percent of the adult population (>18 years) in Jerusalem tested positive for SARS-CoV-2 during 2020 (24,529/605,426) compared to 7.1% of HHUMC HCWs (317/4470), rate ratio 1.75 (95% CI 1.57–1.95), with wide variability (range 0.38–25.0) among different neighborhoods. Of the 30 neighborhoods with more than 50 infected HCWs, 25 showed a higher positivity rate for HCWs compared to the general population. The higher risk of HCWs compared to residents representing the general population in most neighborhoods in Jerusalem may be explained by their behavior in and out of the hospital.

## 1. Introduction

Healthcare workers (HCWs) worldwide have been especially threatened by COVID-19. Hundreds of thousands have contracted the virus, and many have succumbed to the disease [1]. Estimates of the incidence and prevalence of SARS-CoV-2 infection among HCW vary widely with major disease burden [2,3,4,5,6]. COVID-19 infection affected all sectors of the health community to different extents [7].

Personal protective equipment (PPE) can protect HCWs against exposure to respiratory viruses [8]. PPE shortages and re-utilization increase the risks for COVID-19 morbidity [2,9]. Adequate supply and proper PPE use, in conjunction with strict social distancing protocols, have been reported to mitigate and even eliminate SARS-CoV-2 infection [10].

In Israel, the first cases of COVID-19 infection were identified in March 2020. Three consecutive pandemic waves were recorded the following year; the last one for the year 2020 started at end of November. At that time, the number of positive cases in the general population surpassed 300,000, with more than 2500 fatalities [11]. By the end of December 2020—at the height of the third wave—vaccines against SARS-CoV-2 were made available, first to healthcare workers and high-risk groups and later to the general public, with the booster dose available 6 months later.

The first COVID-19 waves demonstrated differential risks for morbidity and mortality by ethnicity. Worldwide, minorities have been found to be especially vulnerable [12]. In Israel, rates of infections and even timing of disease waves varied by subpopulations as well [13]. The Jewish ultra-Orthodox and Arab subpopulations are distinct and are characterized by large family sizes, a younger population, household crowding, lower socio-economic status, and communal religious and ritual observance [14].

Jerusalem is the largest, most diverse city in Israel and includes the largest Jewish, ultra-Orthodox Jewish, and Arab populations nationwide. By the end of 2018, 919,400 persons were listed as Jerusalem residents, and 555,800 were listed as Jewish (60.4%), of whom 223,500 were identified as ultra-Orthodox Jewish (40% of the Jewish and 24% of the total population); the Arab population comprised 346,900 individuals (37.7% of the total population) [15]. Jerusalem in general and the ultra-Orthodox and the Arab populations are notable for a younger population compared to the whole state of Israel [15].

Two of the three major hospitals in Jerusalem are under the auspices of the Hadassah Medical Organization. The two facilities, Ein Karem (EK) and Mount Scopus (MS), include 1044 hospital beds, with an additional 158 beds allocated to COVID-19 patients at the peak of the pandemic. Hadassah Medical Center cared for more than 3000 COVID-19 patients during 2020. The two facilities are generally comparable regarding staffing and the communities they serve. Of note, up to the end of 2021, inpatient COVID-19 wards were exclusive to the EK facility, and cases detected in the MS facility were transferred for further inpatient treatment to EK.

Through the work of the Infection Prevention and Control Unit, contact tracing and epidemiologic investigations were conducted following every case of in-hospital COVID-19 exposure. Early reports stated that healthcare-associated COVID-19 infection was low in Israel as a whole [16] and specifically at Hadassah [17]. Proper HCW protection is multi-layered and includes social distancing guidelines and reinforcing responsible behavior at home, the use of suitable PPE, and the continued instruction on their proper use. In addition to an HCW per demand PCR-based nasopharyngeal screening program, regardless of exposure or symptoms, rapid epidemiologic investigations in cases of contact with COVID-19 cases provided added safety and led to low rates of HCW infection [18,19].

In previous reports, it was unclear whether SARS-CoV-2-positive HCWs were infected at work or outside the hospital. While behaviors such as mask wearing and social distancing can be monitored and even enforced in the hospital, this is not the case when HCWs are at home or in their social settings. Additionally, some HCWs are involved in high-risk aerosol-generating procedures, such as bronchoscopy, airway management, etc. For this reason, we aimed to compare the SARS-CoV-2-positivity rate among HCWs to the rates in the general population in their respective area of residence. Our hypothesis was that there should be no difference in infection rate between HCWs and the general population in their neighborhoods.

## 2. Methods

We assessed the characteristics of Hadassah Medical Center workers, who were positive for SARS-CoV-2 by PCR in a retrospective descriptive study, from 1 March 2020 to 31 December 2020. We included all personnel from the medical, nursing, paramedical, ancillary, and administrative staff. Information about positive cases, including presumed source of infection, were obtained from the hospital’s Unit for Infection Prevention and Control. Basic characteristics including age, gender, occupation, and area of residence were gathered as listed in the hospital’s human resources registries. The proportion of positive HCWs by gender, age subgroups, and occupation were calculated. PCR tests were done in the hospital’s clinical virology laboratory, using various methods throughout the pandemic, including sample pooling, which allowed increased testing capacity with minimal effect on sensitivity and specificity [20].

Cumulative data regarding the incidence and prevalence of SARS-CoV-2-positivity and case characteristics, including age, gender, and dominant ethnic population in Jerusalem’s neighborhoods, were obtained through the Jerusalem District Health Office, Israeli Ministry of Health. Information obtained from the Jerusalem District Health Office did not include communities outside of Jerusalem, and thus, hospital workers not residing in the city of Jerusalem were excluded.

Allocation to neighborhoods for the HCWs group and the general population was in accordance with the records of Jerusalem Health District Office. Neighborhoods without Hadassah workers were excluded as well as all COVID-19 cases under the age of 18 years in the general population. Workers with missing or incomplete addresses were excluded as well (Figure 1).

Rate ratios and 95% confidence intervals of COVID-19 positivity for the hospital staff and the general population in each neighborhood were calculated using WinPepi version 11.65 (Brixton Health). A correction factor of 0.5 was added to the number of positive cases of each neighborhood to assist in rate ratio calculations. Rate ratios were calculated for the population of Jerusalem as a whole, for each neighborhood and for the three main ethnic sub-groups: Jewish, Arabs, and ultra-Orthodox Jewish.

Association of ethnicity with SARS-CoV-2 positivity rate was compared between ethnic groups using the chi-square test and odds ratios and 95% confidence intervals were calculated comparing the ultra-Orthodox with other groups.

Spatial representation of SARS-CoV-2 positivity prevalence for Hadassah workers and neighborhoods was generated, and the absolute numbers of Jerusalem and Hadassah SARS-CoV-2-positive cases were plotted over time.

## 3. Results

HHUMC employs 7760 workers, including 1432 physicians (18%) and 2250 nurses (29%). During the study period, 579 COVID-19-positive workers were reported: 7.5% of the hospitals’ total staff. Their characteristics are described in Table 1.

When stratifying the positive HCWs by age groups, the age group with the highest number of positive cases was 27 to 35 years (204/579, 35.2%). For 18/579 (3.1%), the confirmed case’s age was not listed in the data obtained. The mean and median age of infected workers was 38.1 ± 12.3 and 34 years, respectively; lower quartile age was 28 years, and upper quartile was 47 years. Of all the affected individuals, 327/579 (56%) were female and 252 (43.5%) male. General demographic information for the entire hospitals staff was not available, and thus, ratios were not calculated.

All occupational sectors were affected, and the largest affected subgroup was the nursing staff, comprising 35.4% of all positive cases and 9.1% of the nursing work force. The heterogeneous group of administrative and technical staff included 267 cases: 46.1% of COVID-19 cases in the hospital—and 7.24% cases of a total 3685 workers.

Only 60/579 cases (10.4%) were presumed to be hospital-acquired following the epidemiological investigation. For 52 cases (9%), an in-hospital source was tracked, half of which were acquired from other workers (26/52) and the other half from positive patients (26/52); 123/579 cases (21.2%) were presumed to be acquired from a known source outside of the hospital. However, the source of acquisition was not traced in most cases—396/579 (68.4%) of the total confirmed COVID-19 cases in the hospital staff.

Among Jerusalem residents, during the study period, 51,909 confirmed cases were recorded, 24,529 (47.2%) of whom were above the age of 18 and resided in neighborhoods common to Hadassah workers and were thus included in our study (Table 2). The overall positivity rate for the adult population of the city of Jerusalem was 4%—lower than the 7.5% positivity rate for the hospital staff.

Higher rates of positivity were recorded in the ultra-Orthodox Jewish neighborhoods, comprising 42.5% of all positive cases, although they comprise only 24% of Jerusalem’s total adult population. The other two major groups—the Arab population (35% of total adult residents) and the Jewish population (41% of total adult residents)—had lower positivity rates: 27.5% and 21% of all positive cases, respectively. Among the positive cases, 4488 (8.6%) were not assigned to a specific neighborhood or an ethnic group in the available records and were thus excluded from the analysis. The proportions of PCR-positive cases among the three sub-populations were 7.9%, 3.8%, and 3% of the total ultra-Orthodox, Arab, and Jewish populations, respectively. The odds ratio for positivity was 2.78 for ultra-Orthodox vs. other Jews (95% CI 2.71–2.85), *p* < 0.001, and 2.17 for ultra-Orthodox vs. Arabs (95% CI 2.12–2.22), *p* < 0.001 (chi-square test).

Of the 7760 Hadassah workers, 4470 (57.6%) met inclusion criteria of being registered as Jerusalem residents with complete addresses, divided between 70 neighborhoods and geographical areas. The population of those neighborhoods totaled to 605,426 (94.4%) adult residents (excluded were the population and positive cases in 10 neighborhoods with no Hadassah workers residents, Figure 1). The overall positivity rate ratio for Hadassah workers was 1.75 (95% CI 1.57–1.95), nearly two-fold higher compared to the general Jerusalem adult population.

Out of the Hadassah workers whose domicile was in Jerusalem, 3256/4470 (72.84%) resided in just 40 neighborhoods, which were mainly Jewish. In contrast, 20.44% and 6.71% lived in 14 Arab and 16 ultra-Orthodox neighborhoods, respectively. Rate ratios for SARS-CoV-2 positivity among Hadassah workers relative to the general population in their neighborhood were significantly higher for those residing in the Jewish (2.13, 95% CI 1.84–2.49) and the Arab (3.56, 95% CI 3.05–4.20) neighborhoods but not for those living in the ultra-Orthodox neighborhoods (1.07, 95% CI 0.73–1.64).

The absolute number of positive cases was plotted over time for the entire Jerusalem population and Hadassah workers (Figure 2). Similar analysis by neighborhood was not possible due to the low number of cases per month in each neighborhood.

When subdivided by the 70 Jerusalem neighborhoods, the rate ratios for COVID-19 infection were consistently higher than one, meaning higher positivity in the HCWs group than in their neighbors, in all but nine neighborhoods. In those six Jewish and three ultra-Orthodox neighborhoods (Appendix A), there was no significant difference. Time analysis by neighborhood was unattainable due to the low number of cases per month in each neighborhood.

Out of the 61 neighborhoods with rate ratios higher than one, 36 had a statistically significant increase. All 14 Arab neighborhoods in Jerusalem had statistically significant increased rate ratios for HCWs SARS-CoV-2 acquisition in comparison to their respective control population in contrast to only two (out of 16) Ultra-Orthodox neighborhoods with similar findings.

Figure 3 shows the 30 neighborhoods with more than 50 Hadassah workers. Of these, 25 showed rate ratios for HCWs higher than one (19 of them statistically significant), and five neighborhoods had ratios lower than one (none statistically significant).

A spatial representation for SARS-CoV-2 Positivity for each neighborhood in Hadassah workers is presented in Figure 4.

## 4. Discussion

COVID-19 infection poses a significant hazard to health workers of all sectors and ages. In our study, we observed a high infection rate in HCWs as previously reported [21,22].

Nursing as a profession and female gender have been found previously to be risk factors for COVID-19 infection in some studies [7,23,24] but not in others [25]. Infection by SARS-CoV-2 is associated with poor compliance with infection control guidelines and lack of protective equipment, which are factors not unique to the mentioned subpopulations and do not explain their higher risk. Longer work hours were reported previously [26] to correlate with increased risk and might partially explain such difference due to extended work shifts, fatigue, and longer exposure to COVID-19 inpatients. However, the younger mean age of the nursing staff might imply high-risk behavior outside of the work environment, such as lack of social distancing; yet younger age alone is not a consistent predictor of such behaviors [27].

Hadassah workers had a higher risk for infection relative to adults in the general population in Jerusalem. While only 10% of SARS-CoV-2 acquisition was confirmed as hospital-acquired, in most cases, a source of infection could not be identified (68%) and thus could not be deemed as “hospital-unrelated”. Such data undermine the assumption that the healthcare setting is a relatively safe work environment provided the provision of adequate PPE and other infection control measures. While the use of PPE lowered the risk of SARS-CoV-2 infection [28,29], and shortages were associated with increased risk [9,10], in-hospital infection control programs and adequate PPE supplies might not be enough to eliminate the inherent risk of working with COVID-19 patients. HCW-to-HCW spread was prevalent in the hospital-acquired infection group; failure of social distancing in the work setting is a clear pitfall and an underappreciated risk contributor to the health working community [30]. Similarly, a high positivity rate was seen in non-clinical HCWs in several studies, suggesting that in-hospital transmission is not related directly to patient care [25]. Prompt and early interventions, including environmental modification (e.g., visible signs urging social distancing, virtual meetings, limiting number of personnel in staff- and dining rooms) and early detection and isolation of infected HCWs [5,18] should assist in mitigating such risks.

When subdivided to the three major ethnic and religious groups, HCWs residing in the Arab neighborhoods were found to be the most severely affected. Arabs are an ethnic minority in Israel and Jerusalem, and the observation of a higher rate of infection acquisition in this HCWs subgroup is in alignment with observed vulnerability of HCWs from community minorities in other countries [12,31]. Complex mechanisms probably underlie such increased risk, including out-of-hospital behaviors and distinct socio-demographic characteristics, including low socioeconomic status, household crowding, and communal religious and ritual observance [2,3].

In contrast, workers residing in ultra-Orthodox neighborhoods had lower rates of infection compared to the general Jewish and Arab hospital worker populations, which is inconsistent with the higher-than-average prevalence of COVID-19 infection in this population during the first and second pandemic waves [14,32,33]. The ultra-Orthodox Jewish communities share some characteristics with the Arab minorities, including lower socio-economic status and household crowding [14]. Such differences in risk of infection might be explained by the lower representation of the ultra-Orthodox population in the hospitals’ workforce, impeding the ability to detect a significant difference from the general population, which may have led to imprecise results. Other contributors would be asymptomatic, not-reported infections or test evasion due to stigmatization [34].

Recurrent trends of high infection susceptibility in different minorities highlight the importance of targeted community-specific programs to curb the spread of the current and future pandemics.

No geographical area had significant reduced risk for COVID-19 acquisition in the HCW population in comparison to their respective general population. More alarming, more than half of them (36/70) had statistically significant increased risk for COVID-19 infection for hospital staff (Appendix A).

While the source of many of these infections was not identified, hospital-based infection acquisition cannot be excluded. Patients pose an overt source for SARS-CoV-2 acquisition, and covert HCW-to-HCW spread due to failure of social distancing measures previously led to local outbreaks [30]. Continued research for in-hospital risk factors and improvements to HCWs safety must be regularly undertaken.

Other out-of-hospital community risk factors might be more important than hospital factors in determining HCW’s risk of infection [35]. Education programs must emphasize that the risk also lurks in the seemingly safe community and family environment and that social distancing and compliance with infection control guidelines must continue outside of the workplace.

Our study, although retrospective, encompassed a large study population and comparison group. Limitations include its retrospective nature and the reliance on workers’ self-reported area of residence, which might not have been up to date. The two Hadassah facilities are generally comparable in regard of staff and the communities they serve although the referral of SARS-CoV-2-confirmed patients to the larger campus might have altered exposure risk. Widespread surveillance and non-obligatory staff screening recommendations in Hadassah might have contributed to the higher infection rate observed relative to the community although the wide availability of non-symptom restricted, whole population, and on-demand testing implemented in the community increases the validity of the findings. Obviously, the community samples were tested in several different laboratories, while most HCWs’ tests were performed in the hospital’s central virology laboratory; however, the same validated commercial kits were used in all locations, so this difference should not limit the results. Lack of representation of first on-site responders, HCWs in the community settings, and the low participation of the ultra-Orthodox community in the medical sector workforce hinders the ability to generalize the findings to all sectors. Lastly, these findings do not reflect the effects of vaccinations against SARS-CoV-2, which were available only in the last days of 2020.

## 5. Conclusions

Our study shows increased risk for infection of hospital workers across most ethnic subgroups when compared to infection rates in their area of residence. These findings are inconsistent with the popular notion that abundance of PPE and their proper use will eliminate any risk for COVID-19 acquisition for HCWs. This increased risk might reflect differences in HCWs behavior when treating COVID-19 patients and when in contact with their colleagues or, more concerning, the failure of the health system to protect its workers. Further studies are needed to define the precise pathways for infection acquisition and ways to increase protection in hospital settings for the current and future pandemics.

## Figures and Tables

**Figure 1 ijerph-19-12951-f001:**
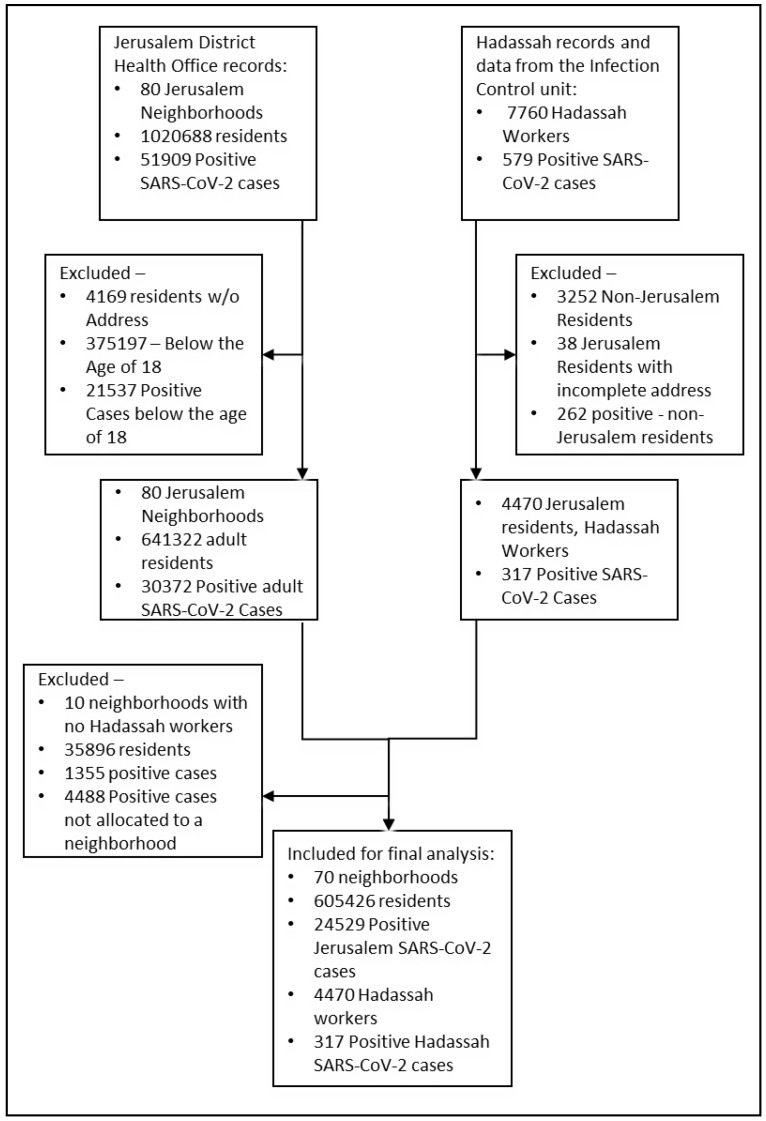
Flow diagram of study population inclusions and exclusions.

**Figure 2 ijerph-19-12951-f002:**
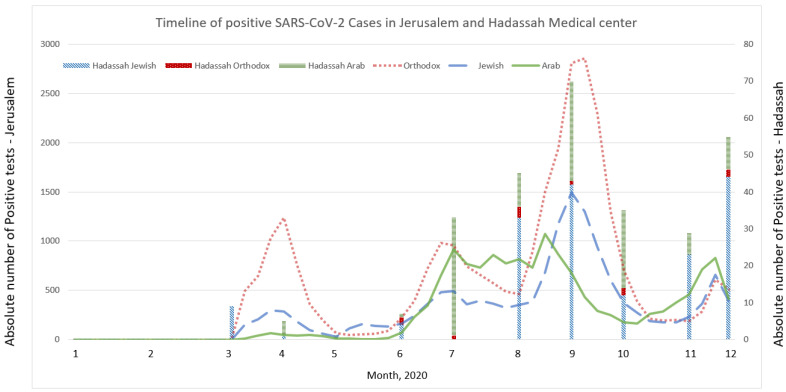
Absolute number of SARS-CoV-2-positive tests over 2020 in Jerusalem and Hadassah medical center. Colored lines represent the Jerusalem population, and stacked columns represent the Hadassah hospital group.

**Figure 3 ijerph-19-12951-f003:**
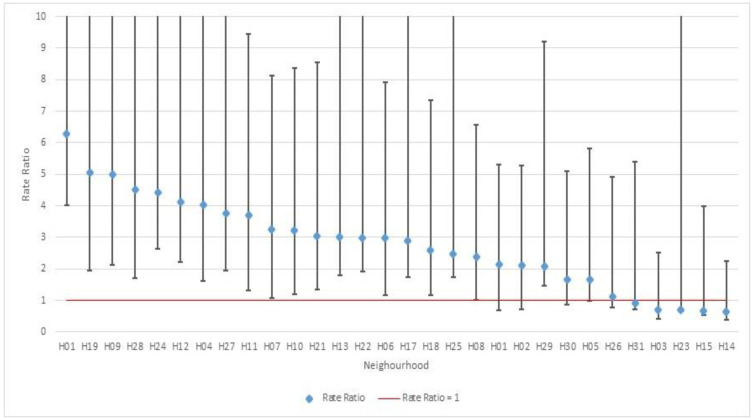
Comparison of the rate of positive healthcare workers to the rate of positivity in the general public per neighborhood. Rate Ratios and confidence intervals for the 30 largest neighborhoods in Jerusalem in descending order.

**Figure 4 ijerph-19-12951-f004:**
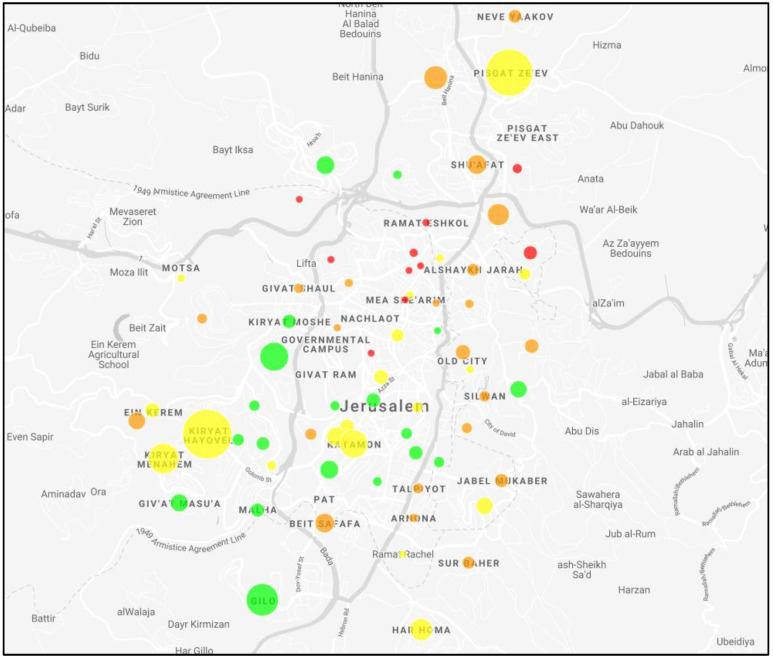
Heat map of SARS-CoV-2 prevalence in Hadassah workers in Jerusalem. Circle size corresponds with number of Hadassah workers and color with SARS-CoV-2 prevalence: green < 5%, yellow 5–10%, orange 10–20%, red > 20%.

**Table 1 ijerph-19-12951-t001:** Demographic characteristics of SARS-CoV-2-positive healthcare workers, March–December 2020 (n = 579).

**Age Groups**			
**Age Group, years**	**Positive Cases, N (%) ^a^**		
**18–26**	95 (16.4%)		
**27–35**	204 (35.2%)		
**36–44**	90 (15.5%)		
**45–53**	87 (15.0%)		
**54–62**	63 (10.9%)		
**63–71**	22 (3.8%)		
**Not specified**	18 (3.1%)		
**Work Sectors**			
**Sector**	**Positive Cases, N (% of total staff)**	**Positive Cases, N (% of total cases)**	
**All**	579/7760 (7.5%)	579 (100.0%)	
**Medical**	80/1432 (5.5%)	80/579 (13.8%)	
**Nursing**	205/2250 (9.1%)	205/579 (35.4%)	
**Paramedical**	27/384 (7.0%)	27/579 (4.6%)	
**Administrative**	267/3694 (7.2%)	267/579 (46.1%)	
**Source of Infection according to epidemiologic investigation**			
**Hospital Acquired**	60/579 (10.3%)		
**Community Acquired**	123/579 (21.2%)		
**Unknown**	396/579 (68.3%)		
**Ethnic subgroups in healthcare workers residing in Jerusalem**			
	**Positive cases out of subgroup, N (%)**	**Positive out of total positive workers N (%)**	**Total Number of Workers N (%)**
**Total**	317/4470 (7.0%)	317 (100.0%)	**4470 (100.0%)**
**Jewish**	164/3256 (5.0%)	164/317 (51.7%)	**3256/4470 (72.8%)**
**Arab**	131/914 (14.3%)	131/317 (41.3%)	**914/4470 (20.4%)**
**Ultra-Orthodox Jewish**	22/300 (7.3%)	22/317 (6.9%)	**300/4470 (6.7%)**

^a^ Percentage of positive cases in each age group out of total positive workers.

**Table 2 ijerph-19-12951-t002:** Jerusalem SARS-CoV-2-positivity rates by ethnic sub-populations.

	Positive Cases N (%)	Total Population	% Positivity
**Total Population (Including Children)**			
**Total population**	51,909 (100.0%)	1,020,688 (100.0%)	5.0%
**Jewish (%)**	11,034 (21.2%)	366,446 (35.9%)	3.0%
**Arab (%)**	14,303 (27.5%)	372,750 (36.5%)	3.8%
**Ultra-Orthodox Jewish (%)**	22,084 (42.5%)	277,317 (27.1%)	7.9%
**Adult Population (Above 18 Years Only)**			
**Total adult population**	24,529 (100.0%)	605,426 (100.0%)	4.0%
**Jewish (%)**	5829 (23.7%)	247,212 (40.8%)	2.3%
**Arab (%)**	8164 (33.2%)	203,309 (33.5%)	4.0%
**Ultra-Orthodox Jewish (%)**	10,536 (42.9%)	154,905 (25.5%)	6.8%

## Data Availability

The data presented in this study are available on request from the corresponding author. The data are not publicly available due to institutional policy.

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
