# Peer review of "Are Healthcare Workers Infected with SARS-CoV-2 at Home or at Work? A Comparative Prevalence Study"

_ijerph, 2022, doi:10.3390/ijerph191912951_

Round 1
Reviewer 1 Report
-The data presented in this work does not support the conclusions. This manuscript is full of claims without showing any evidence; for example, the authors use terms like “abundance of PPE and their proper use” - there is no data provided about PPE use, etc.
- The study is designed poorly. The motivation behind the main hypothesis of this study is not explained in the manuscript. Why is it essential to compare infection rates between health workers and the general population considering health workers are part of the general population? For health workers, there are almost equal chances of getting infected in the hospital or outside of their work location. Did the authors have data about where the health workers were infected? If not - comparing spatial infection rates is not logical.
- None of the data is analyzed using statistical tests. The authors should consider highlighting the significant differences while comparing the data between the two populations.
Author Response
– the study rationale is presented in the introduction. During the early covid-19 waves when vaccines were unavailable, the question of whether health care workers were infected at work or at home was highly relevant (for example doi:10.1001/jamanetworkopen.2020.9666 and others). In Israel PPE was widely available in hospitals and there was no rationing of PPEs in the hospital environment, meaning that HCW could be protected at work.
- Regarding the main hypothesis – statistical analysis was certainly performs and reflected in the 95% confidence intervals of the rate ratios- These provide a much more comprehensive view of the precision of the RR estimates and differences between neighborhoods than would multiple p values.
Reviewer 2 Report
Are healthcare workers infected with SARS-CoV-2 at home or at work? A comparative prevalence study
The work is based on statistical analyses regarding the infectivity rates among healthcare workers that comprised of different 3 sub-ethnic populations. It represents an interesting piece of work as whole that might be helpful in the future to plan and minimize the infection risks among the said working personals as well as an informative and pre-emptive source of action for the government and healthcare officials. Since the study was carried out b/w Mar - Dec of 2020 where vaccines were only available after this duration, it can be speculated that the data presented has nothing in correlation with the vaccination program at a later stage. However, what seems intriguing is that the nutritional, health, social behavior like drinking, smoking etc., and other physical aspects of living standards of the healthcare workers is not mentioned in detail that certainly would have an impact on the statistical data analyses. For some reasons the low socioeconomic and living status of some Arab neighborhoods and ultra-orthodox Jewish communities are highlighted but the fact is that in majority of such other reports, this discriminatory factor remained to a minimum, meaning that mostly the medium-high class population were also affected and hence the healthcare workers. Such factual scenarios needs to be addressed as the COVID-19 pandemic through aerosol droplets has almost spared none of the population communities regardless of their ethnic background or socioeconomic status rather their nutritional, health and other social habits.
Author Response
We thank the reviewer for these comments. Unfortunately, we do not have any data regarding any other behaviors such as smoking as suggested.
Additionally, many healthcare workers do not belong to the higher socioeconomic groups, such as cleaning and technical workers, and even students who work in part-time jobs.
Reviewer 3 Report
This manuscript was written well and attractive in the field of communicable diseases, especially in Israel's urban, which have distinct communities and religions from other countries.
Minor concerns.
1. Please add the version of the WinPepi program.
2. Figure S1. It seems too confusing because the lines and bar are similar colours.
Please revise this figure again and suggest adding a sign into the line such as square, circle and triangle to make it clear, especially for readers who printout on monochrome paper.
Optional.
1. Introduction. You can add more risky activities such as "aerosol-generating procedures (e.g. nasopharyngeal swab, intubation, bronchoscopy, sputum induction...)".
This manuscript content is fair for publication and also distinct from other papers because this manuscript describes the Jewish community that cannot be seen in other countries.
However, the tables and visualisation in this manuscript could be improved for an easier read.
1. Should be added deep detail about risk factors (smoking, close contact, lifestyle, risk behaviours, wearing a facemask properly...) in the HCWs group and regenerate tables and visualisation.
2. The principal point of this manuscript is "comparative prevalence". The prevalence in normal people must be deep in detail than the current version.
3. An age subgroup in table 1.
3.1. Positive cases should be reported by cases/total (%column) that would be easily understood.
3.2 Jerusalem population should be reported by cases/total (%column) too.
4. Table 1. Gender, Work Sector, Source of Infection according to epidemiologic investigation and Ethnic subgroups in healthcare workers residing in Jerusalem should be separated in the new table.
5. Tables must add a description.
Author Response
This manuscript was written well and attractive in the field of communicable diseases, especially in Israel's urban, which have distinct communities and religions from other countries.
We thank the reviewer for these comments.
Minor concerns.
- Please add the version of the WinPepi program.
We added version number.
- Figure S1. It seems too confusing because the lines and bar are similar colours. Please revise this figure again and suggest adding a sign into the line such as square, circle and triangle to make it clear, especially for readers who printout on monochrome paper.
We have revised the colors and added dashed lines to make the figure easier for black-and-white reading.
Optional.
- Introduction. You can add more risky activities such as "aerosol-generating procedures (e.g. nasopharyngeal swab, intubation, bronchoscopy, sputum induction...)".
We added this important point to the introduction.
This manuscript content is fair for publication and also distinct from other papers because this manuscript describes the Jewish community that cannot be seen in other countries.
However, the tables and visualisation in this manuscript could be improved for an easier read.
- Should be added deep detail about risk factors (smoking, close contact, lifestyle, risk behaviours, wearing a facemask properly...) in the HCWs group and regenerate tables and visualisation.
We do not have details on such activities, as we collected retrospective data on positive PCR tests and did not interview the HCWs.
- The principal point of this manuscript is "comparative prevalence". The prevalence in normal people must be deep in detail than the current version.
The comparison is between HCWs and residents of same neighbourhoods, not for the entire population. This comparison is detailed in table S1.
- An age subgroup in table 1.
To avoid confusion we edited table 1 and removed the general population age groups, as we do not have data regarding the age of positive non-HCWs.
3.1. Positive cases should be reported by cases/total (%column) that would be easily understood.
3.2 Jerusalem population should be reported by cases/total (%column) too.
The age distribution of all workers was not available, therefore we cannot present as suggested. We did edit table 2 to be more comprehensible.
- Table 1. Gender, Work Sector, Source of Infection according to epidemiologic investigation and Ethnic subgroups in healthcare workers residing in Jerusalem should be separated in the new table.
We separated every category of subgroups and removed some data that was duplicated in the Results section of the manuscript.
- Tables must add a description.
We added description for every table as requested.
Round 2
Reviewer 1 Report
As I mentioned before, please incorporate which tests you used to check the significance of each comparison and what were the p values. Just mentioning it in one line is not good practice.
Author Response
As I mentioned before, please incorporate which tests you used to check the significance of each comparison and what were the p values. Just mentioning it in one line is not good practice.
We thank the reviewer for these comments. We added the required data in the methods and results sections.
Reviewer 2 Report
The authors have made effort to improve the manuscript. The manuscript may now be accepted for publication.
Author Response
Thank you
Reviewer 3 Report
Comments.
1. The table is incomplete and unattractive. Please adjust it again to make it more clear before the next process.
2. Suggest moving Figure S1 to Figure 1 (included in the manuscript) to make it a more attractive paper, also Figure S2.
3. Please revise Table S1 again.
3.1 What is the type of the "(CI)"?
Is it 95%CI, 99%CI or so on...
Please add this point to make it more clear.
3.2 Please considered a "significant figure" in the xx% CI and correct the typos;
3.2.1 For example, revise from (2.01 - 5.1) to (2.01 - 5.1x),
3.2.2 For example, (13.66 - 17.) to (13.66 - 17.xx).
Author Response
- The table is incomplete and unattractive. Please adjust it again to make it more clear before the next process.
We edited Table 1 again, making it complete and nice, including fixing some typos.
- Suggest moving Figure S1 to Figure 1 (included in the manuscript) to make it a more attractive paper, also Figure S2.
We thank the reviewer for this suggestion, we moved both figures to the manuscript as Figures 3 and 4.
- Please revise Table S1 again
We revised Table S1 as suggested.
3.1 What is the type of the "(CI)"?
Is it 95%CI, 99%CI or so on...
Please add this point to make it more clear.
We used 95% confidence intervals. We added this clarification where it was missing.
3.2 Please considered a "significant figure" in the xx% CI and correct the typos.
We changed all fractions to be consistent and fixed some minor typos.